# A Damping-Tunable Snap System: From Dissipative Hyperchaos to Conservative Chaos

**DOI:** 10.3390/e24010121

**Published:** 2022-01-13

**Authors:** Patinya Ketthong, Banlue Srisuchinwong

**Affiliations:** School of Information, Computer, and Communication Technology, Sirindhorn International Institute of Technology, Thammasat University, Pathum Thani 12120, Thailand; patinya.ket@gmail.com

**Keywords:** adjustable attractor dimension, conservative chaos, dissipative hyperchaos, FPAA, hyperjerk, tunable damping, snap

## Abstract

A hyperjerk system described by a single fourth-order ordinary differential equation of the form $\ddddot{x}=f\left(\dddot{x},\ddot{x},\dot{x},x\right)$ has been referred to as a snap system. A damping-tunable snap system, capable of an adjustable attractor dimension ($D_{L}$) ranging from dissipative hyperchaos ($D_{L}<4$) to conservative chaos ($D_{L}=4$), is presented for the first time, in particular not only in a snap system, but also in a four-dimensional (4D) system. Such an attractor dimension is adjustable by nonlinear damping of a relatively simple quadratic function of the form $Ax^{2}$, easily tunable by a single parameter *A*. The proposed snap system is practically implemented and verified by the reconfigurable circuits of field programmable analog arrays (FPAAs).

## 1. Introduction

Studies of chaotic systems have received great attention due to many practical applications in science and technology [1,2,3]. Typically, a three-dimensional (3D) chaotic system is expressed by a set of three coupled first-order ordinary differential equations (ODEs), whereas a four-dimensional (4D) chaotic system is expressed by a set of four coupled first-order ODEs.

On the other hand, four successive time derivatives of displacement (*x*), for example, are known as velocity ($\dot{x}$), acceleration ($\ddot{x}$), jerk ($\dddot{x}$) [4,5,6], and snap ($\ddddot{x}$ or hyperjerk) [5], whereas a time derivative higher than the third is known as hyperjerk [7]. A chaotic jerk system is described by a single third-order ODE of the form $\dddot{x}=f\left(\ddot{x},\dot{x},x\right)$, whereas a chaotic snap (or hyperjerk) system is described by a single fourth-order ODE of the form $\ddddot{x}=f\left(\dddot{x},\ddot{x},\dot{x},x\right)$ [7,8].

Chaos is typically measured by Lyapunov exponents (LEs) and a Lyapunov dimension ($D_{L}$). The latter is alternatively known as a Kaplan–Yorke dimension ($D_{KY}$) or an attractor dimension [5]. The largest Lyapunov exponent (LLE) measures chaoticity [9], i.e., how much a system is sensitive to the initial conditions. A system will exhibit chaos if one LE is positive, but will exhibit hyperchaos if at least two LEs are positive [10]. On the contrary, the Lyapunov dimension or attractor dimension measures the complexity (strangeness) [9,11] of an attractor and is based on the calculation of all LEs. In addition, the negative sum of all LEs represents damping (α), the dissipation of energy from oscillations, of the form $\alpha =-\sum L_{i}$, where $L_{i}$ is the *i*-th LE, and therefore, damping plays an essential role not only in oscillations, but also in the value of an attractor dimension [5,12].

The attractor dimension ($D_{L}$) also classifies whether chaos (or hyperchaos) in an *n*-dimensional, or *n*-th-order, system is dissipative ($D_{L}$ < *n*) or conservative ($D_{L}$ = *n*). Although not all undamped systems can exhibit chaos, conservative chaos (or hyperchaos) is however found in an undamped system where the average damping along the trajectories is zero (α=0) [5]. Conservative chaos (or hyperchaos) is therefore of special interest [13], but appears to be the minority of reported chaotic (or hyperchaotic) systems, whereas dissipative chaos (or hyperchaos) appears to be the majority. Although a transition between dissipative and conservative chaos is possible through adjustable damping for an adjustable attractor dimension, such a transition is however relatively rare. In particular, transitions ranging from dissipative hyperchaos to conservative chaos have never been reported.

In 3D chaotic flows, existing transitions from dissipative chaos ($D_{L}<3$) to conservative chaos ($D_{L}=3$) have been found in, for example, a damped, forced, pendulum [5] and in [14], of which the attractor dimension is adjustable to the maximum, i.e., $D_{L}\leq 3$. On the contrary, such transitions to the maximum attractor dimension $D_{L}\leq 4$ have never been found in 4D or fourth-order (snap or hyperjerk) chaotic flows, although an attempt in a simple damping-tunable single-transistor-based chaotic snap circuit [12] has demonstrated its maximized attractor dimension (but $D_{L}\neq 4$ as $D_{L}=3.28156$) at its minimized damping (but α≠0 as α≅1).

Recently, field programmable analog arrays (FPAAs) [15], which offer reconfigurable circuits for analog signal processing, have been employed in implementing chaotic circuits and systems [16,17,18]. The advantages of FPAAs include that a mathematical model, such as a chaotic model, can be practically implemented by configurable circuits of FPAAs with ease, flexibility, reliability, and robustness. Such advantages greatly reduce the time and costs of a prototype.

In this paper, a new damping-tunable snap (hyperjerk) system capable of exhibiting conservative chaos, dissipative chaos, and dissipative hyperchaos is proposed. In addition, transitions ranging from dissipative hyperchaos to conservative chaos are demonstrated for the first time, especially not only in a snap system, but also in a 4D system. Its dynamical properties were investigated. Circuits for the new snap system were implemented through the use of FPAAs, which allowed us to verify its dynamical properties.

## 2. The Proposed Snap System

### 2.1. The General Form of a Snap System

Many simple hyperchaotic snap systems [7,19,20,21,22,23,24,25,26] have been presented in a single fourth-order ODE, which can be written in a general form as:(1)$$\ddddot{x}=-\alpha \dddot{x}-a_{1}\ddot{x}-a_{2}\dot{x}-a_{3}x-a_{4}g\left(\ddot{x},\dot{x},x\right)$$
where α is a damping coefficient [5,12] of the form:(2)$$\alpha =a_{0}f\left(x\right)$$$a_{0},a_{1},a_{2},a_{3},a_{4}$ are parameters, and f(x) and $g(\ddot{x},\dot{x},x)$ are nonlinear functions. As a result, the damping coefficient in (Equation 2) is a nonlinear damping coefficient. Table 1 compares such existing hyperchaotic snap systems [7,19,20,21,22,23,24,25,26] and shows that they do not exhibit conservative chaos, where ($A_{1}$ to $A_{8}$) are tuning parameters and ($B_{0}$ to $B_{7}$), $C_{8}$, ($E_{1}$ to $E_{8}$), ($F_{5}$, $F_{6}$, $F_{8}$), and ($G_{6}$, $G_{8}$) are constants, all of which are not equal to one. Note that $B_{9}$ is described in Section 2.4.

### 2.2. A New Damping-Tunable Snap System

Further studies through individual simulations showed that existing systems [7,19,20,21,22,23,24,25] based on (Equation 1) continue to exhibit hyperchaos for positive values of $a_{2}$, whereas the existing nonlinear function $f\left(x\right)=x^{4}$ can be alternatively replaced by a new simpler nonlinear function $f\left(x\right)=x^{2}$. In particular, additional studies revealed that conservative chaos is possible if the damping coefficient α=0 and the parameter $a_{2}=0$. Such studies led to a new single fourth-order ODE of the proposed snap system of the form:(3)$$\ddddot{x}=-A\left(x^{2}\dddot{x}+\dot{x}\right)-B\ddot{x}-\left|\dot{x}\right|-x$$
where *A* is a tuning parameter, B=2.65 is a constant, $f\left(x\right)=x^{2}$, and $g\left(\ddot{x},\dot{x},x\right)=\left|\dot{x}\right|$, as also included in Table 1. It can be noticed from Table 1 that the nonlinear function f(x) in (Equation 3) is a quadratic function of the form $x^{2}$, whereas f(x) in existing systems [7,19,20,21,22,23,24,25] is a quartic function of the form $x^{4}$. Although the system in [26] has f(x)=1, the function $g(\ddot{x},\dot{x},x)$ consists of many trigonometric functions that lead to difficulties in circuit implementation.

Based on (Equation 2) and Table 1, the damping coefficient in (Equation 3) is of the form:(4)$$\alpha =Ax^{2}$$By tuning the parameter *A*, as will be shown later, the proposed snap system in (Equation 3) can exhibit not only dissipative hyperchaos, but also conservative chaos.

Equivalently, the damped snap system in (Equation 3) can be rewritten as a 4D damped dynamical system in a set of four coupled first-order ODEs of the form:(5)$$\left\{\begin{array}{cc} \dot{x} & =y\\ \dot{y} & =z\\ \dot{z} & =w\\ \dot{w} & =-A(wx^{2}+y)-2.65z-|y|-x \end{array}\right.$$
where the state variables are defined as:(6)$$\left[\begin{array}{cccc} x & y & z & w \end{array}\right]=\left[\begin{array}{cccc} x & \dot{x} & \ddot{x} & \dddot{x} \end{array}\right]$$By substituting $\dot{x}=0,\dot{y}=0,\dot{z}=0$ and $\dot{w}=0$ in (Equation 5), the corresponding equilibrium point in (Equation 5) is located at $E_{1}=\left(x_{0},y_{0},z_{0},w_{0}\right)=\left(0,0,0,0\right)$.

### 2.3. Snap-Based Dissipative Hyperchaos

The 4D damped dynamical system in (Equation 5) was numerically simulated by using the adaptive fourth-order Runge–Kutta (RK4) integration [27] with an adaptive step size Δt≤0.01. Figure 1a,b, respectively, shows the Lyapunov dimension ($D_{L}$) and the spectrum of LEs ($L_{1},L_{2},L_{3},L_{4}$), ordered from large to small values, of (Equation 5) versus the parameter *A* from 0–0.5. The calculations of the spectrum of LE are based on the algorithm proposed by Wolf et al. [28]. An initial condition is (x,y,z,w)=(0.1,0.1,0.1,0.1). To ensure that chaos and hyperchaos are neither transients nor numerical artifacts, the calculations of the Lyapunov dimension and LEs follow the orbit for a sufficiently large time of $t=10^{8}$.

As shown in Figure 1b, where A≠0 and $D_{L}<4$, the damped snap system in (Equation 3) can exhibit not only snap-based dissipative chaos, but also snap-based dissipative hyperchaos. The latter exists for A∈(0.22,0.35).

For system stability, the corresponding Jacobian matrix is written as:(7)$$J_{1}=\left[\begin{array}{cccc} 0 & 1 & 0 & 0\\ 0 & 0 & 1 & 0\\ 0 & 0 & 0 & 1\\ -2Awx & -A-sgn(y) & -2.65 & -Ax^{2} \end{array}\right]$$For example, at A=0.275, where the dissipative hyperchaos exists, the eigenvalues are $\lambda_{1,2}=0.0759\pm 1.4996i$ and $\lambda_{3,4}=-0.0759\pm 0.6617i$ at the equilibrium point $E_{1}=\left(0,0,0,0\right)$, and therefore, $E_{1}$ is a spiral saddle equilibrium with index 2.

For the dissipative chaos and hyperchaos in the damped snap systems in (Equation 3), the (positive) damping coefficient in (Equation 4) refers to the (negative) rate of a phase space expansion [5,12,28] of the form:(8)$$\alpha =-\nabla \cdot F=-\langle Tr\left(J\right)\rangle =-\sum_{i=1}^{4}L_{i}$$
where *F* is the flow in (Equation 5), ∇·F is the time-averaged divergence of the flow *F*, 〈Tr(J)〉 is the trace of the Jacobian matrix $J_{1}$ in (Equation 7) averaged along the trajectory, and $L_{i}$ is the *i*-th LE. As a result, the damped snap system in (Equation 3) is a dissipative system.

### 2.4. Snap-Based Conservative Chaos

In addition to the snap-based dissipative hyperchaos, the proposed snap system in (Equation 3) is capable of exhibiting snap-based conservative chaos at A=0 where $D_{L}=4$, as shown in Figure 1a, and therefore, the damped snap system in (Equation 3) is reduced to an undamped snap system of the form:(9)$$\ddddot{x}=-2.65\ddot{x}-\left|\dot{x}\right|-x$$
where the damping coefficient in (Equation 4) is α=0. The resulting undamped snap system in (Equation 9) appears to be in a similar form as an existing undamped snap system of the form [8]:(10)$$\ddddot{x}=-B_{9}\ddot{x}\pm \left|\dot{x}\right|-x$$
where $B_{9}=2.525$, as also included in Table 1. This is the first time, especially in a snap system and a 4D system, that transitions between conservative chaos in (Equation 9) and dissipative hyperchaos in (Equation 3) can be realizable.

Equivalently, the undamped snap system in (Equation 9) can be rewritten as a 4D undamped dynamical system of the form:(11)$$\left\{\begin{array}{cc} \dot{x} & =y\\ \dot{y} & =z\\ \dot{z} & =w\\ \dot{w} & =-2.65z-|y|-x \end{array}\right.$$The corresponding Jacobian matrix is of the form:(12)$$J_{2}=\left[\begin{array}{cccc} 0 & 1 & 0 & 0\\ 0 & 0 & 1 & 0\\ 0 & 0 & 0 & 1\\ -1 & -sgn(y) & -2.65 & 0 \end{array}\right]$$At A=0, where the conservative chaos exists, the eigenvalues are $\lambda_{1,2}=0\pm 1.4919i$ and $\lambda_{3,4}=0\pm 0.6703i$ at the equilibrium point $E_{2}=\left(0,0,0,0\right)$, and therefore, $E_{2}$ is a nonhyperbolic equilibrium, of which at least one eigenvalue has a zero real part. In particular, all eigenvalues of $E_{2}$ have no real part. Such eigenvalues lie on the imaginary axis with the absence of real parts for conservative chaos.

For the conservative chaos in the undamped systems in (Equation 9), the damping coefficient in (Equation 8) becomes zero, where *F* is the flow in (Equation 11), and 〈Tr(J)〉 is the trace of the Jacobian matrix $J_{2}$ in (Equation 12) averaged along the trajectory. As a result, the phase space of the systems in (Equation 9) is neither compressible nor expandable, and therefore, the undamped snap system in (Equation 9) is a conservative system.

A non-geometric integrator such as the RK4 integrator may affect the behavior of chaotic systems, especially of conservative systems, as non-geometric integrators may not fully preserve the properties of the continuous prototype in discrete calculations [29]. In an attempt to avoid possible dissipation appearing from such numerical methods, both the trace of the Jacobian matrix averaged along the trajectories and the summation of all LEs in (8) have been closely monitored to ensure the conservation of energy. The former always results in zero, whereas the latter is taken as zero if $|\sum_{i}L_{i}|<0.001$, enabling the estimated uncertainty of less than $0.001-|\sum_{i}L_{i}|$.

### 2.5. Tunable Damping: From Dissipative Hyperchaos to Conservative Chaos

Substituting the damping coefficient in (Equation 4) into (Equation 8) yields:(13)$$\alpha =Ax^{2}=-\sum_{i=1}^{4}L_{i}$$It can be observed from (Equation 13) that changes in the parameter *A* result in not only changes in the Lyapunov dimension (attractor dimension $D_{L}$), as shown in Figure 1a, but also changes in the damping coefficient (α), as shown in Figure 2. In particular, Figure 3 depicts the Lyapunov dimension (attractor dimension $D_{L}$) in Figure 1a versus the damping coefficient (α) in Figure 2.

In a similar manner to the damping coefficient (α) in (Equation 8), it is evident from Figure 3 that a decrease in damping, e.g., from α= 0.2655 to 0.0, allows a phase space expansion with the tendency of an increase in the Lyapunov dimension (i.e. attractor dimension), e.g., from $D_{L}=$ 1.0 to 4.0. In particular, the minimized damping at α = 0.0 yields the maximized attractor dimension at $D_{L}$ = 4.0. Such an expansion demonstrates transitions, ranging from dissipative hyperchaos to conservative chaos, for the first time, in particular in a snap system and in a 4D system.

### 2.6. Multistability and Coexisting Attractor

Figure 4 illustrates the basins of attraction of (Equation 11) in red and purple areas of conservative chaos and periodic oscillations, respectively, on an (x,y) plane, where z=w=0. The equilibrium point is a blue dot at (x,y)=(0,0). The system (Equation 11) therefore exhibits multistability and two coexisting oscillations. Figure 5 shows two examples of asymmetric coexisting attractors, of which the initial conditions are (0.01,0,0,0) and (−0.01,0,0,0) for conservative chaos and periodic oscillations, respectively.

## 3. FPAA-Based Circuit Implementation

It is customary that the behavior of a newly proposed chaotic model can be verified by a proof through its analog circuit implementation such that properties such as numerical trajectories of the simulated model can be directly compared to oscilloscope-trace trajectories of its real experimental circuit for verification. In particular, it is increasingly customary that field programmable analog arrays (FPAAs) are used for such an analog circuit implementation.

FPAAs are programmable integrated circuits that enable flexible and rapid prototyping of analog circuits using configurable analog modules (CAMs). In addition, FPAAs provide efficient and economical solutions to design analog dynamical systems with increased reliability. Two (or four) FPAA chips, each of which is AN231E04, are on an Anadigm DualApex (or QuadApex) Development Board and are programmed and designed for circuits by AnadigmDesigner2 [15].

To avoid possible confusions, it should be emphasized that an FPAA is totally different from an field programmable gate array (FPGA). An FPAA is based on programmable analog building blocks, e.g., integrators, filters, and switched-capacitor circuits, whereas an FPGA is based on configurable logic modules or look-up tables and does not involve analog building blocks. In particular, FPAAs are fully programmable analog circuits, whereas FPGAs are programmable digital circuits. Although an FPGA may require a procedure such as a linearization method to convert differential equations to the binary or digital terms for the implementation on an FPGA [30], an FPAA does not at all require this.

Typically, the design procedure of FPAAs can be divided into two parts. The first part involves mathematical models, numerical simulations, and scaling processes. The latter may be required if the amplitude of the chaotic signals is larger than the supply voltage of the FPAAs. The second part involves the system being able to be modeled by CAMs, which are programmed into FPAA chips. If there are any mismatches between the numerical simulations from the first part and the experimental results from the second part, such mismatches may be eliminated by modifying the parameters inside the CAMs.

As an integrator of FPAAs is scaled by a constant k=1/τ, where τ is the time constant of the integrator, e.g., if τ=0.4 ms, then k=0.0025 (1/us), the scaling processes of the snap system in (Equation 3) are required. In other words, the system in (Equation 5) is scaled to be:(14)$$\left\{\begin{array}{cc} \dot{x} & =k_{1}y\\ \dot{y} & =k_{2}z\\ \dot{z} & =k_{3}w\\ \dot{w} & =k_{4}[-A\left(wx^{2}+y\right)-2.65z-|y|-x] \end{array}\right.$$
where the scaling factors are $k_{1}=k_{2}=k_{3}=0.0025$ (1/us) and $k_{4}=0.00306$ (1/us).

The scaled system in (Equation 14) is implemented by the configurable circuits of the first chip (FPAA1) for the dissipative hyperchaos with A≠0, as shown in Figure 6, whereas (Equation 14) is implemented by the configurable circuits of the second chip (FPAA2) for the conservative chaos with A=0, as shown in Figure 7. Each configurable circuit consists of four integrator blocks and a full-wave rectifier block. The latter is for the absolute function. The state variables x,y,z, and *w* are the outputs of individual integrator blocks.

For dissipative hyperchaos, the nonlinear damping coefficient (α) is implemented by additional circuits using two multiplier blocks and a summing block, as shown in Figure 6. For values of the scaling factors in (Equation 14), Figure 8 and Figure 9 [15] show the parameters and time constants of CAMs for the first (FPAA1) and second (FPAA2) chips, respectively. In addition, clock-A of the individual components is configured by default at 250 kHz, whereas clock-B is additionally required by the multipliers to be 16-times that of clock-A at 4 MHz.

## 4. Numerical and Experimental Results

For the proposed snap system, the numerical trajectories of the simulated model in Section 2 and oscilloscope-trace trajectories of FPAA-based real experiments in Section 3 are compared in this section. The oscilloscope-trace trajectories are readily available from the FPAAs and, therefore, can be directly measured by an analog oscilloscope (e.g., GW-Instek GOS-620) without the need for extra ADC or DAC circuits.

On the one hand, at A=0.275, for the snap-based dissipative hyperchaos, the numerical trajectories of the attractors are illustrated in Figure 10a–d, whereas the corresponding FPAA-based oscilloscope traces are shown in Figure 10e–h, on the (*x*, $\dot{x}$), (*x*, $\ddot{x}$), (*x*, $\dddot{x}$), and ($\ddot{x}$, $\dddot{x}$) planes, respectively.

On the other hand, at A=0, for the snap-based conservative chaos, the numerical trajectories of the attractors are illustrated in Figure 11a–d, whereas the corresponding FPAA-based oscilloscope traces are shown in Figure 11e–h, on the (*x*, $\dot{x}$), (*x*, $\ddot{x}$), (*x*, $\dddot{x}$), and ($\ddot{x}$, $\dddot{x}$) planes, respectively. It can be noticed from Figure 10 and Figure 11 that both the numerical results and FPAA-based experimental results are in good agreement.

## 5. Conclusions

A new damping-adjustable fourth-order hyperjerk (snap) system was proposed and capable of tunable nonlinear damping for an adjustable attractor dimension ($D_{L}$), resulting in transitions ranging from dissipative hyperchaos with $D_{L}<4$ to conservative chaos with $D_{L}=4$. Such transitions were demonstrated for the first time, in particular in either a snap system or a 4D system. The tunable nonlinear damping was based on a relatively simple quadratic function of the form $Ax^{2}$, conveniently adjustable by a single tuning parameter *A*. The experimental results were based on the reconfigurable circuits of FPAAs and corresponded to the numerical simulations.

## Figures and Tables

**Figure 1 entropy-24-00121-f001:**
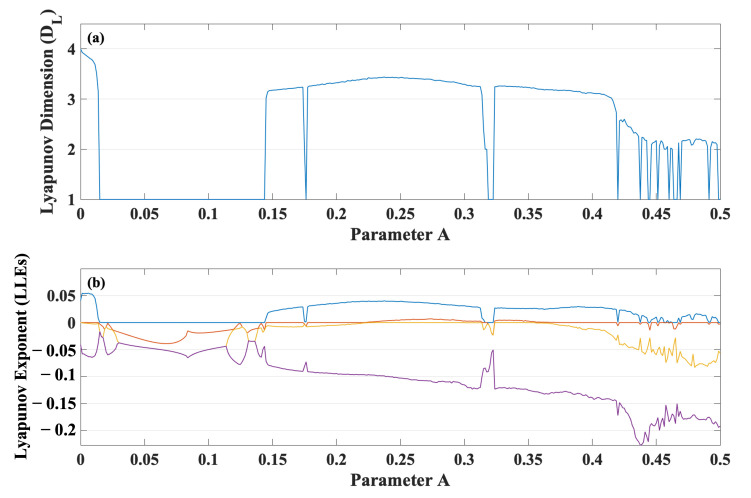
(**a**) The Lyapunov dimension ($D_{L}$). (**b**) The spectrum of LEs ($L_{1},L_{2},L_{3},L_{4}$), ordered from large to small values, of (Equation 3) versus *A*.

**Figure 2 entropy-24-00121-f002:**
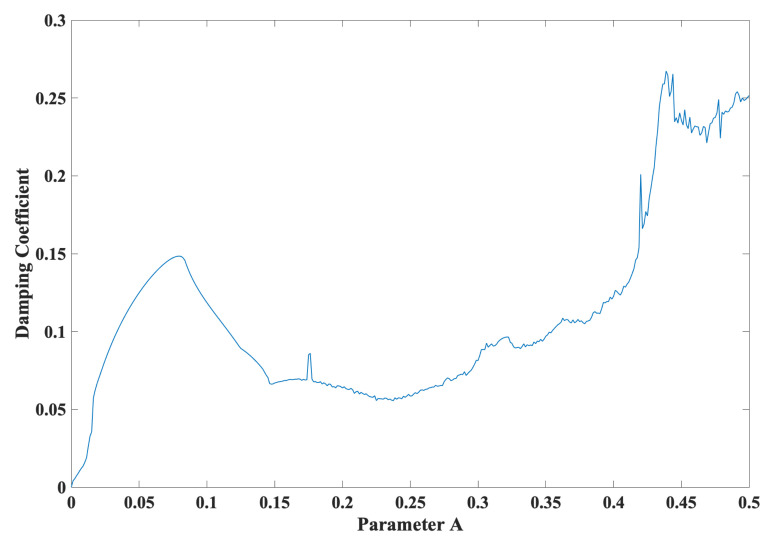
The damping coefficient (α) versus the parameter *A*.

**Figure 3 entropy-24-00121-f003:**
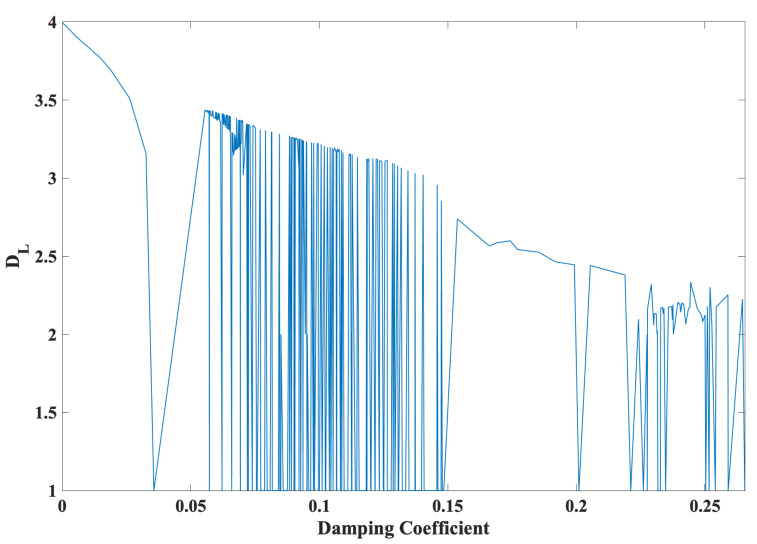
A decrease in damping, from α= 0.2655 to 0.0, allows a phase space expansion with the tendency of an increase in the attractor dimension, from $D_{L}=$ 1.0 to 4.0.

**Figure 4 entropy-24-00121-f004:**
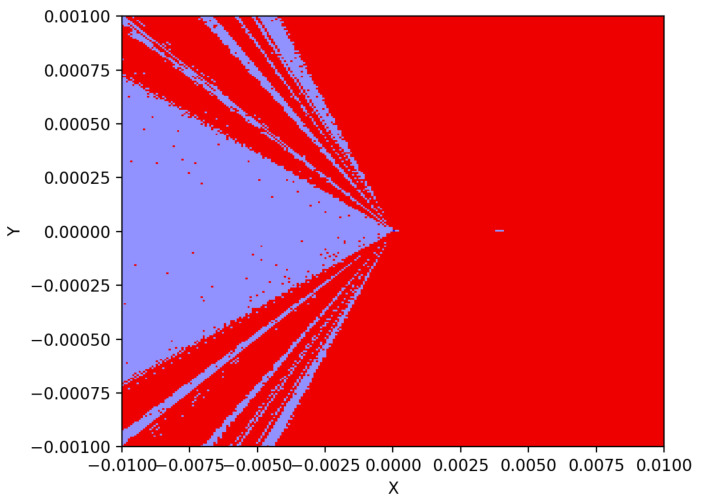
Basins of attraction of (Equation 11) in red and purple areas of conservative chaos and periodic oscillations, respectively, on an (x,y) plane where z=w=0. The blue dot is the equilibrium point at the origin. The system therefore exhibits multistability and coexisting attractors.

**Figure 5 entropy-24-00121-f005:**
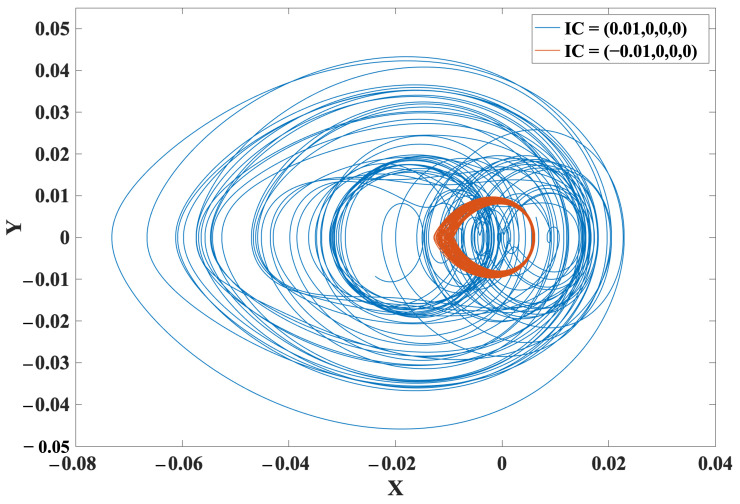
Two examples of coexisting attractors of (Equation 11), of which the initial conditions are (0.01,0,0,0) and (−0.01,0,0,0) for conservative chaos and periodic oscillations, respectively.

**Figure 6 entropy-24-00121-f006:**
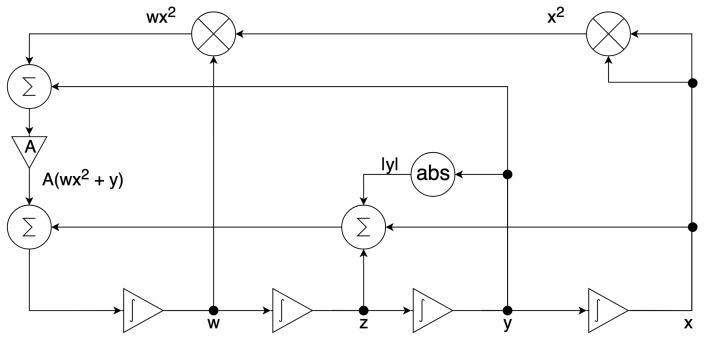
Circuit implementation of the scaled system in (Equation 14) where A≠0 for dissipative hyperchaos using the first chip (FPAA1).

**Figure 7 entropy-24-00121-f007:**
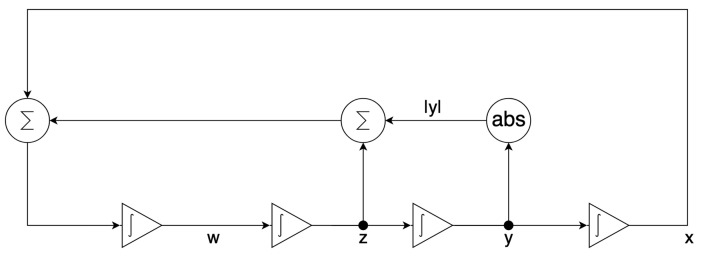
Circuit implementation of the scaled system in (Equation 14) where A=0 for conservative chaos using the second chip (FPAA2).

**Figure 8 entropy-24-00121-f008:**
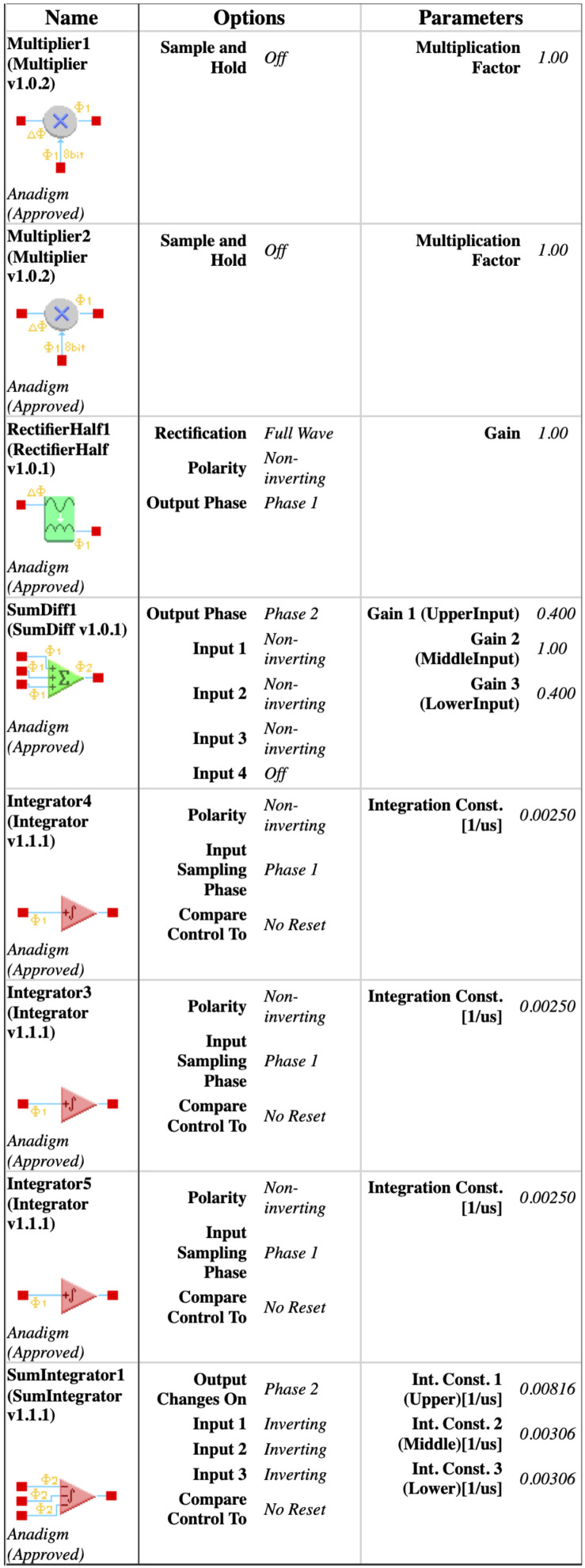
Parameters of the configurable analog modules [15] of the first chip (FPAA1) for dissipative hyperchaos.

**Figure 9 entropy-24-00121-f009:**
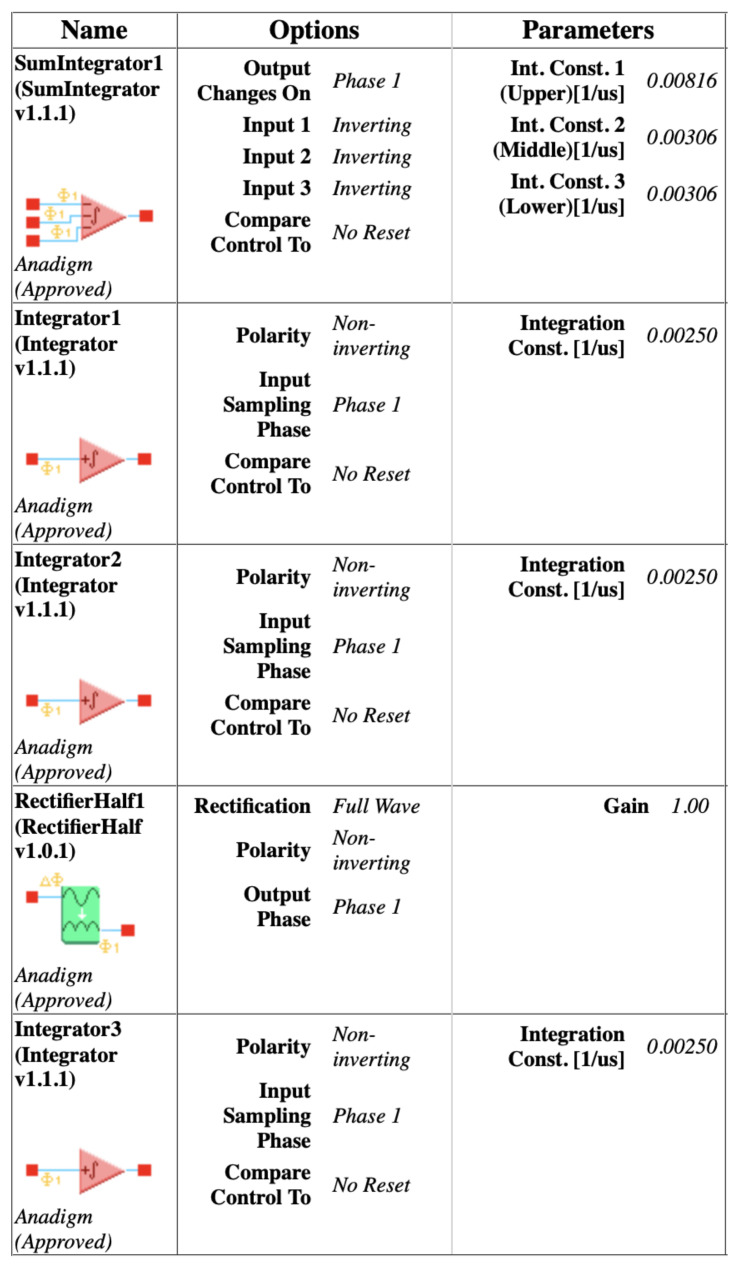
Parameters of the configurable analog modules [15] of the second chip (FPAA2) for conservative chaos.

**Figure 10 entropy-24-00121-f010:**
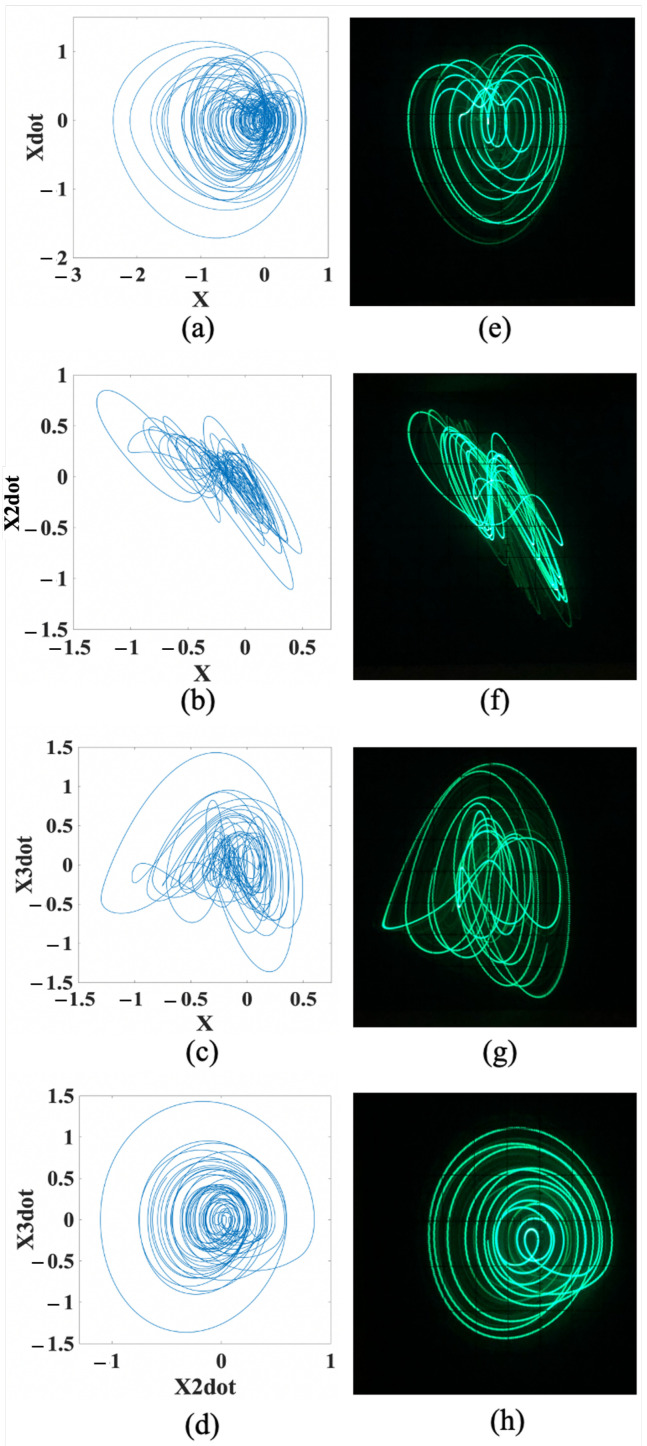
Attractors of snap-based dissipative hyperchaos on the (*x*, $\dot{x}$), (*x*, $\ddot{x}$), (*x*, $\dddot{x}$), and ($\ddot{x}$, $\dddot{x}$) planes. (**a**–**d**) Numerical trajectories, respectively. (**e**–**h**) FPAA-based oscilloscope traces, respectively.

**Figure 11 entropy-24-00121-f011:**
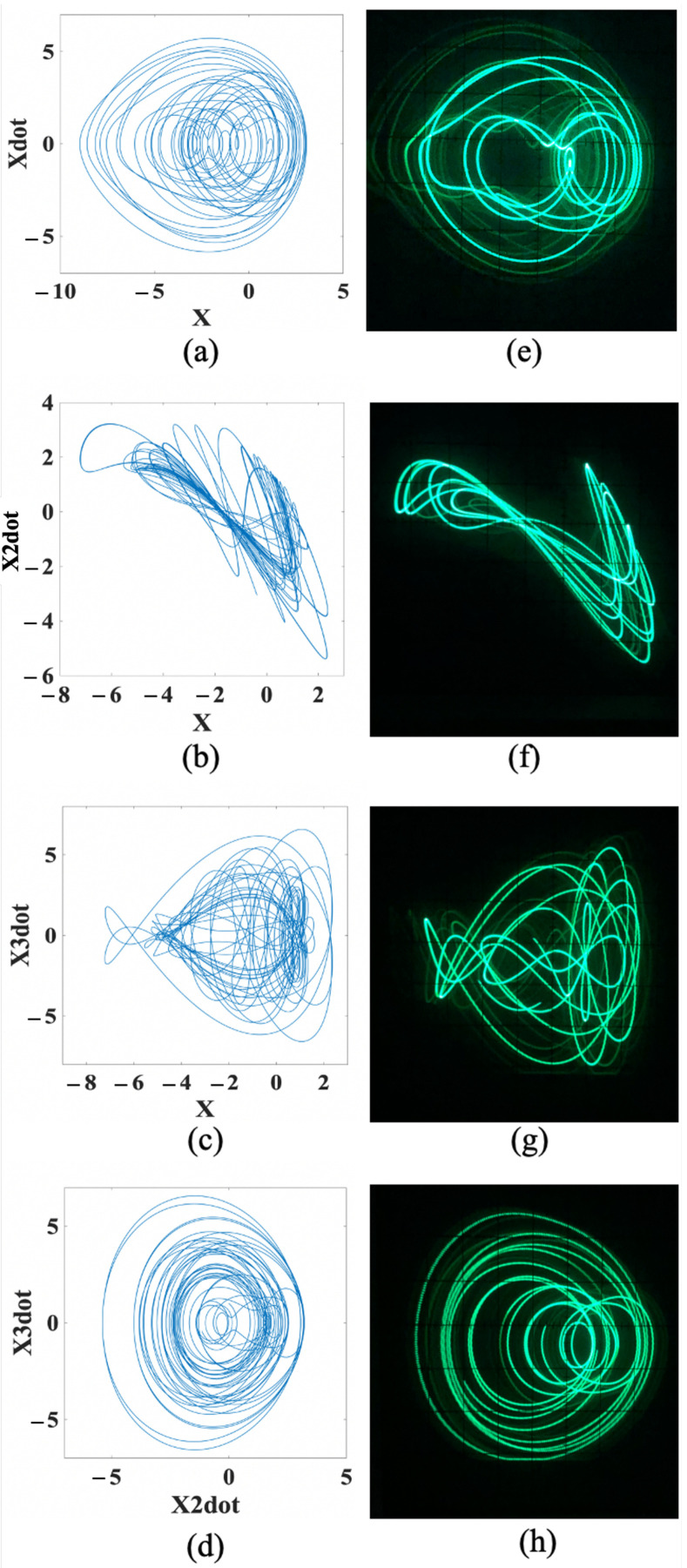
Attractors of snap-based conservative chaos on the (*x*, $\dot{x}$), (*x*, $\ddot{x}$), (*x*, $\dddot{x}$), and ($\ddot{x}$, $\dddot{x}$) planes. (**a**–**d**) Numerical trajectories, respectively. (**e**–**h**) FPAA-based oscilloscope traces, respectively.

**Table 1 entropy-24-00121-t001:** Comparisons of hyperchaotic snap systems.

References	f(x)	$a_{0}$	$a_{1}$	$a_{2}$	$a_{3}$	$a_{4}$	$g(\ddot{x},\dot{x},x)$	Numbers of Terms	Dissipative Hyperchaos	Conservative Chaos
[7]	$x^{4}$	1	$B_{0}$	1	1	0	−	4	🗸	×
[19]	$x^{4}$	$A_{1}$	$B_{1}$	1	1	$E_{1}$	$\sinh(\dot{x})$	5	🗸	×
[20]	$x^{4}$	$A_{2}$	$B_{2}$	1	1	$E_{2}$	$x^{2}$	5	🗸	×
[21]	$x^{4}$	$A_{3}$	$B_{3}$	1	1	$E_{3}$	|x|	5	🗸	×
[22]	$x^{4}$	$A_{4}$	$B_{4}$	1	1	$E_{4}$	$|\dot{x}|$	5	🗸	×
[23]	$x^{4}$	$A_{5}$	$B_{5}$	1	1	$E_{5}$	$|\dot{x}|+F_{5}{\dot{x}}^{2}$	6	🗸	×
[24]	$x^{4}$	$A_{6}$	$B_{6}$	1	1	$E_{6}$	$|\dot{x}|+F_{6}{\dot{x}}^{2}+G_{6}\left|\ddot{x}\right|$	7	🗸	×
[25]	$x^{4}$	$A_{7}$	$B_{7}$	1	1	$E_{7}$	${\ddot{x}}^{3}$	5	🗸	×
[26]	1	$A_{8}$	−1	$C_{8}$	0	$E_{8}$	$\tan\left(\ddot{x}\right)-F_{8}\sin\left(x\right)\ddot{x}+G_{8}\mathrm{sgn}\left(x\right)$	6	🗸	×
[8]	−	0	$B_{9}$	0	1	1	$|\dot{x}|$	3	×	🗸
This paper	$x^{2}$	*A*	*B*	*A*	1	1	$|\dot{x}|$	5	🗸	🗸

## Data Availability

Not applicable.

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
