# Peer review of "A Damping-Tunable Snap System: From Dissipative Hyperchaos to Conservative Chaos"

_entropy, 2022, doi:10.3390/e24010121_

Round 1
Reviewer 1 Report
The authors has proposed a snap system that can exhibit dissipative as well as conservative chaos by tuning a single parameter. Calculating Lyapunov exponents they also show that there are parameter regimes where the system can show chaos or hyper chaos. They also demonstrate multi stability and coexisting attractors in the model. The model is clearly introduced with clear demonstration of various interesting properties of chaos. Further more, their numerical simulations are supported by experiments on circuits of Field Programmable Analog Arrays. So, I recommend the publication of the article.
Reviewer 2 Report
The paper presents a novel hyperjerk chaotic system and studies its implementation through programmable analog arrays.
The paper is exceptionally well-written and I am in view that it can be published on Entropy, pending some minor corrections.
I have uploaded a scanned version of the paper carrying some hand-written notes to improve the literal presentation.
